# Jet modifications from colour rope formation in dense systems of non-parallel strings

Christian Bierlich, Smita Chakraborty⋆, Gösta Gustafson, and Leif Lönnblad

Department of Astronomy and Theoretical Physics, Sölvegatan 14A, S-223 62 Lund, Sweden
⋆ smita.chakraborty@thep.lu.se,
† MCNET-22-02, LU-TP 22-09

March 7, 2022

## Abstract

We revisit our rope model for string fragmentation that has been shown to give a reasonable description of strangeness and baryon enhancement in high-multiplicity pp events at the LHC. A key feature of the model is that the enhancement is driven by the increased string tension due to strings overlapping in dense systems. By introducing an improved space–time picture for the overlap between fragmenting strings, where also non-parallel strings are properly taken into account, we are now able to investigate the enhancement both in jets and in the underlying event in a consistent way.

# 1  Introduction

One of the most characteristic features of Quark–Gluon Plasma (QGP) formation in heavy ion (AA) collisions, is that of so–called "jet quenching" [1]. In heavy ion collisions, jet quenching is mainly seen in energy loss or dispersion effects, manifest as, for example, suppression of high $p_\perp$ particle yields, with respect to scaled proton–proton (pp) case [2] or the suppression of away-side jets in central collisions [3]. With the higher energies available at LHC, the phenomenon has also been explored using Z bosons plus jets, where the Z decaying to leptons is used as an unaffected probe, to gauge the effect on the jet traversing the QGP [4].

Several experimental signatures for QGP production have, however, also been observed in high multiplicity pp collisions, including strangeness enhancement [5] and long-range multi-particle correlations [6], more commonly known as "collective flow". Jet quenching effects have so far not been observed in small systems (pp or proton–ion, p$A$), which begs the question if jet modification phenomena are completely absent in small systems, or if the correct way to look for it has just not been established [7]. One obvious reason for the difficulty, is that it is not possible, like in $AA$ collisions, to look at differences when comparing to similar measurements for pp collisions. Comparisons with theoretical expectations are also difficult, as the expected effects from quenching in small systems are very small, and the signal is strongly affected by (uncertain) effects from initial state radiation. It should also be mentioned that most theoretical descriptions of jet quenching assumes that the jet is formed in a deconfined "bath" of free partons (*i.e.* the QGP), which may not be appropriate in small systems, where at most a few droplets would form. This includes approaches such as QGP-modified splitting kernels [8] at high virtualities, coupled with shower modifications by transport theory [9, 10] at lower ones, but also approaches like the one offered by JEWEL [11], where partonic rescattering off medium partons are combined with the Landau–Pomeranchuk–Migdal effect [12].

In a series of papers we have demonstrated, that collective flow and enhancement of strangeness and baryons, can be reproduced in high multiplicity pp events as a result of string-string interaction, when the infinitely thin string is generalized to a confining colour fluxtube, similar to a vortex line in a superconductor [13]. As discussed in ref. [14], models of string interactions offers a novel and convenient framework for studying jet modifications in small systems, as they are implemented in the general purpose Monte Carlo event generator PYTHIA, which allows the user to generate realistic collision events, with the effects switched "on" or "off". The study of jet modification effects does therefore not need to rely on a (non-existing) reference system.

The aim of this paper is therefore to look at possible effects of jet modification via increased strangeness and baryon numbers in jets. A very important tool is here the method developed in ref. [15], to account for the interaction between strings which are not parallel to each other. This was not possible in earlier versions of string-string interaction, but is naturally very important for handling the interaction between string pieces connected to a jet and strings in the underlying event.

The remainder of this paper is organised as follows. In section 2, we recap the Lund string hadronization framework, taking into account the transverse extension of strings, and discuss how the string tension increase when such strings overlap, leading to strangeness and baryon enhancement. Then we present the parallel frame and our updated rope model in section 3. In section 4 we investigate how the average string tension varies as a function of multiplicity and transverse momentum, and then investigate the observable modifications the updated rope model predicts for jets and the underlying event in pp collisions at the LHC, before we present our conclusions in section 5.

## 2 String hadronization and colour fluxtubes

In this section we will briefly introduce relevant parts of the Lund string hadronization model, building up to the rope hadronization model used for the model results. For more detailed reviews on Lund strings, we refer the reader to the large body of existing literature. The original papers deal mainly with hadronization of a single straight string [16, 17]. Gluons were introduced as 'kinks' on a string in refs. [18, 19]. Somewhat dated reviews are presented in refs. [20, 21], and a number of recent papers on Lund strings present the model in a more modern context [15, 22–24], including our original paper on rope hadronization [25].

The Lund string is a "massless relativistic string" (or a "Nambu-Goto string"). Such a string has no transverse extension, and it also has no longitudinal momentum, which implies that it is boost invariant. [1] This may be a good approximation for a linear colour fluxtube, where the width is not important. In section 2.2 we will discuss going beyond this approximation.

### 2.1 Lund string hadronization

**Hadronization of a straight string**

We first look at a single, straight string stretched between a quark and an anti-quark. The string can break via $q\bar{q}$ pair creation, in a process which can be regarded as a tunneling process as discussed in ref. [27]. For a single quark species the production probability is given by

$$\frac{\mathrm{d}\mathcal{P}}{\mathrm{d}^2 p_\perp} \propto \kappa \exp\left(-\frac{\pi\mu_\perp^2}{\kappa}\right). \tag{1}$$

Here $\mu_\perp^2 = \mu^2 + p_\perp^2$ is the *quark* squared transverse mass. The exponential conveniently factorizes, leaving separate expressions for selection of mass and $p_\perp$ to be used in the Monte Carlo event generator. With $\kappa \approx 1$ GeV/fm, this result implies that strange quarks are suppressed by roughly a factor 0.3 relative to a u- or d-quark (and that the probability to produce a c-quark with this mechanism is $\sim 10^{-11}$). It also means that the quarks are produced with an average $p_\perp \sim 250$ MeV, independent of its flavour.

When the quarks and antiquarks from neighbouring breakups combine to mesons, their momenta can be calculated as an iterative process. The hadrons are here "peeled off" one at a time, each taking a fraction ($z$) of the *remaining* light-cone momentum ($p^\pm = E \pm p_z$) along the positive or negative light-cone respectively. The probability for a given $z$-value is here given by

$$f(z) \propto \frac{(1-z)^a}{z} \exp(-bm_\perp^2/z). \tag{2}$$

Here $m_\perp$ is the transverse mass of the *meson*, and the two parameters $a$ and $b$ are to be determined by tuning to data from e$^+$e$^-$ collisions. In principle the $a$–parameter could depend on the quark species, but in default PYTHIA (the Monash tune) it is the same for strange and non-strange quarks. Baryon–antibaryon pairs can be produced via production of a diquark–antidiquark pair, and in this case the $a$-parameter has to be modified. The parameter $b$ must, however, be universal.

An important consequence of eq. (2) is the probability distribution in proper time ($\tau$) for string breakup vertices. Expressed in terms of the quantity $\Gamma = (\kappa\tau)^2$, the distribution is given by:

$$\mathcal{P}(\Gamma)\mathrm{d}\Gamma \propto \Gamma^a \exp(-b\Gamma)\mathrm{d}\Gamma. \tag{3}$$

---

[1]For the kinematics of such a string see ref. [26].

The breakup-time is an important ingredient for string interactions, as the hadronization time sets an upper limit on the available time for strings to push each other and form ropes. As such, hadronization of a system of interacting strings will *not* happen when the system has reached equilibrium, but will be cut off when the string hadronizes. For strings hadronizing early, one can then imagine a mixed phase of strings and hadrons, before the transition to a pure hadron cascade. In this paper we consider only the effect of string interactions, and leave the interplay with the hadronic cascade for a future paper. We note, however, that a full hadronic cascade has recently been implemented in PYTHIA [28, 29], revealing only minor effects in proton collisions. Typical values for $a$, $b$ and $\kappa$ give an average breakup time of around 1.5 fm. This can not be identified as the hadronization time (or freeze-out time). This could equally well be interpreted as the time when the quark and the antiquark meet for the first time. In addition the breakup times fluctuate, and each string will hadronize at different times.

**Gluons and non-straight strings**

An essential component in the Lund hadronization model is that a gluon is treated as a point-like "kink" on the string, carrying energy and momentum. A gluon carries both colour and anti-colour, and the string can be stretched from a quark, via a set of colour-ordered gluons, to an anti-quark (or alternatively in a closed loop of colour-ordered gluons).

When a gluon has lost its energy, the momentum-carrying kink is split in two corners, moving with the speed of light but carrying no momentum, stretching a new straight string piece between them. When two such corners meet, they can "bounce off"; the string connecting them then disappears, but a new one is "born". In a pp collision a typical string will contain several gluons, connected by string pieces which are stretched out, may disappear and then be replaced by new string pieces. All these string pieces move transversely in different directions, but at any time the string consists of a set of straight pieces. For a description of how such a string hadronizes, we refer to refs. [15, 30]. The interaction between strings with several non-parallel pieces is discussed in section 3.

## 2.2   Strings as colour fluxtubes

The description of a confining colour field by an infinitely thin string is necessarily an approximation, relevant only when the result is insensitive to the width. In high multiplicity events this is no longer the case, and the strings have to be treated as colour fluxtubes, with a non-zero width. We here first discuss the properties of a single fluxtube, and then the interaction between two or more parallel fluxtubes. The generalization to non-parallel fluxtubes is presented in section 3.

### 2.2.1   A single fluxtube

The simplest model for a QCD fluxtube is the MIT bag model [31]. Here a homogenous longitudinal colour-electric field is kept inside a tube by the pressure from the vacuum condensate. An improved description is obtained in lattice calculations. A common method is here to use the method of Abelian projections, proposed by 't Hooft [32], which is based on partial gauge fixing. The result of these calculations show that the field is dominated by a longitudinal colour-electric field, surrounded by a transverse colour-magnetic current in the confining vacuum condensate [33, 34]. This picture is very similar to the confinement of the magnetic field in a vortex line in a superconductor (with electric and magnetic fields interchanged, see *e.g.* ref. [35]).

As observed in ref. [15] the measured shape of the colour electric field obtained in ref. [36] is well approximated by a Gaussian distribution:

$$E(\rho) = E_0 \exp\left(-\rho^2/2R^2\right), \tag{4}$$

where $\rho$ is the transverse distance in cylinder coordinates. The width of a fluxtube is difficult to estimate in lattice calculations, as it is naturally given in lattice units, see *e.g.* ref. [37]. It is often estimated to be around 0.5 fm.

The field density in eq. (4) is related to the string tension through

$$\int d^2\rho\, E^2(\rho)/2 = \pi E_0^2 R^2 = g\kappa, \tag{5}$$

where $g$ is the fraction of the total energy of the string associated with the colour electric field. We expect $g$ to be of the order $1/2$, which is the value obtained in the bag model, where the energy in the field and the expelled condensate are of equal size. For a further discussion of the vacuum condensate and colour fluxtubes we refer to ref. [38] and references therein.

### 2.2.2  Interacting parallel fluxtubes

High multiplicity collisions will give a high density of fluxtubes, with a corresponding high energy density. In ref. [15] we discussed the collective effects expected from the initial expansion, and in this paper we will concentrate on the effects of rope hadronization, and in particular study the production of strange hadrons. Here we first restate our treatment of interaction between parallel fluxtubes, presented in ref. [25]. How this can be generalized to a general situation with non-parallel fluxtubes will be discussed in section 3 below.

**Rope formation**

For two overlapping parallel fluxtubes, separated by a transverse distance $\delta$, we get from eq. (4) the interaction energy of the field

$$\int d^2\rho\, (\mathbf{E}_1(\rho) + \mathbf{E}_2(\rho))^2/2 - 2\int d^2\rho\, E^2(\rho)/2 = \int d^2\rho\, \mathbf{E}_1(\rho)\cdot\mathbf{E}_2(\rho) = 2\pi E_0^2 R^2 e^{-\delta^2/4R^2}. \tag{6}$$

Such a system will expand transversely, and if it does not hadronize before, it will reach equilibrium, where the energy density corresponds to the free energy density in the vacuum condensate.

The expression in eq. (6) does not include the surface energy for the combined flux tube. In the bag model this is zero, and in equilibrium the transverse area will be doubled, and the interaction energy will be zero. For a vortex line in a dual QCD superconductor, it depends on the properties of the superconductor, but also here the interaction energy will be much reduced at the time of hadronization. It will then be necessary to go beyond the Abelian approximation. For two fluxtubes stretched by quarks, the two quarks can either form a colour sextet or an anti-triplet, and with more fluxtubes also higher multiplets are possible. Here lattice calculations show that a set of overlapping strings form a "rope", with a tension proportional to the second Casimir operator for the colour multiplet at the end of the rope [39].

Biro, Nielsen, and Knoll pointed out [40] that if a rope is formed by a number of strings with random charges, they add up as a random walk in colour space. This implies that the net colour grows as the square root of the number of strings. A rope stretched by $m$ colour charges and $n$ anti-charges can then form a colour multiplet characterised by two numbers $p$ and $q$, such that an arbitrary state, by a rotation in colour space, can be transformed into a state with $p$ coherent colours (*e.g.* red) and $q$ coherent anti-colours (*e.g.* anti-blue), such that

the colour and the anti-colour do not form a colour singlet. Such a multiplet is denoted $\{p, q\}$, and we always have $p \le m$ and $q \le n$.

For any such multiplet we can write down the number of states, *i.e.* the multiplicity[2] of the multiplet:

$$N = \frac{1}{2}(p + 1)(q + 1)(p + q + 2). \tag{7}$$

As mentioned above, the total tension of such a rope is proportional to the second Casimir operator for the multiplet, which gives

$$\kappa^{\{p,q\}} = \frac{C_2(p,q)}{C_2(1,0)}\kappa^{\{1,0\}} = \frac{1}{4}\left(p^2 + pq + q^2 + 3p + 3q\right)\kappa^{\{1,0\}}, \tag{8}$$

where $\kappa^{\{1,0\}} \equiv \kappa$ is the tension in a single string.

In the PYTHIA treatment used here, there are, however, other effects also addressing string coherence effects. Importantly, parts of this colour summation is in an approximate way treated by "colour reconnection". As a simple example we can look at two anti-parallel strings with triplet–anti-triplet pairs in each end. These can either form an octet or a singlet, with probabilities 8/9 and 1/9 respectively. Here the octet (denoted $\{1, 1\}$) gives

$$\kappa^{\{1,1\}} = \kappa \cdot C_2^{\{1,1\}}/C_2^{\{1,0\}} = 9\kappa/4. \tag{9}$$

The singlet ($\{0, 0\}$), with no string at all, gives $\kappa^{\{0,0\}} = 0$.

The colour reconnection process in a situation with several strings can be related to an expansion in powers of $1/N_c$, as discussed in refs. [41, 42].

For the special case of $N_c = 3$ there is also a different kind of reconnection. For a rope formed by two parallel strings, the two triplets in one end can give either a sextet or an anti-triplet (and a corresponding anti-sextet or triplet in the other end) with probabilities 2/3 and 1/3 respectively. For the latter we simply have just a single string.

The two original colour triplets are connected in a "junction", and such a reconnection can be particularly important for baryon production. This possibility is not implemented in the present version of our Monte Carlo, but will be included in future work. We note that for an arbitrary number of colours, the corresponding situation is only obtained when $N_c - 1$ colour charges combine to one anti-colour charge. The junction formation with three strings does therefore, for $N_c \ne 3$, correspond to a configuration where $N_c$ strings are connected, which cannot be directly interpreted as a $1/N_c$ correction.

We will in the following adopt a picture where the process of string (rope) fragmentation follows after a process of colour reconnections, and that this will leave the system in a state with $p$ parallel and $q$ anti-parallel strings forming a coherent multiplet $\{p, q\}$.

**Rope hadronization**

A rope specified by the multiplet $\{p, q\}$, can break via a succession of single $q\bar{q}$ productions, through the tunnelling mechanism in eq. (1). In each step a multiplet $\{p, q\}$ is changed to either $\{p - 1, q\}$ or $\{p, q - 1\}$. It is here important to note that *the tunneling is not determined by the total tension in the rope, but by the energy released, determined by the reduction in the tension* caused by the production of the new $q\bar{q}$ pair. Hence, we get from eq. (8) an effective string tension, when the field goes from $\{p + 1, q\}$ to $\{p, q\}$, given by

$$\kappa_{\text{eff}} = \kappa^{\{p+1,q\}} - \kappa^{\{p,q\}} = \frac{2p + q + 4}{4}\kappa. \tag{10}$$

---

[2]The multiplicity provides the standard nomenclature for multiplets, where $N = 1$ is called "singlet", $N = 3$ is called "triplet", $N = 6$ is called "sextet" etc. We will here, when necessary, use the slightly more verbose notation $\{p, q\}$, which allows one to distinguish between *e.g.* a triplet and an anti-triplet.

The consequence of this picture is that we can treat the rope fragmentation as the sequential decay of the individual strings forming the rope, much in the same way as an everyday rope would break thread by thread. Technically it means that we can use the normal string fragmentation procedure in PYTHIA8, with the modification that we in each break-up change the fragmentation parameters according to the effective string tension calculated from the overlap of neighbouring strings. The changes to these parameter explained in detail in ref. [25], and are for reference also listed in appendix A. The changes are somewhat convoluted, since most of the parameters only indirectly depend on the string tension, but the main effect easily seen in eq. (1), namely that an increased string tension will increase the probability of strange quarks and diquarks relative to light quarks in the string breakup.

## 3   Rope hadronization with non-parallel strings

Our previous work on rope formation [25] relied on the assumption that strings in high energy hadron collisions can be assumed to be approximately parallel to each other and to the beam axes. This prevented a detailed investigation of possible effects in hard jets, especially those traversing the dense environment of an *AA* collision. In our recent work on the shoving model [15] we found a remedy where the interaction between any pair of strings can be studied in a special Lorentz frame, even if they are not parallel to each other or to the beam. We call it "the parallel frame", and it can be shown that any pair of straight string pieces can be transformed into such a frame, where they will always lie in parallel planes.

Below we will use this parallel frame to calculate the increased string tension in the rope formation of arbitrarily complex string configurations.

### 3.1   The *parallel frame* formalism

In the previous rope implementation [25], the way to determine if any two string pieces are overlapping was to boost them to their common centre-of-mass frame and here measure the distance between them at a given space-time point of break-up. This was done in a fairly crude way, not really taking into account that the two string pieces typically cannot be considered to be parallel in this frame. In general there is no frame where two arbitrary string pieces can be considered to be exactly parallel, but in the *parallel frame* introduced in ref. [15] it can be shown that any two string pieces will always be stretched out in parallel planes in a symmetric way. This works for all pairs of string pieces, even if one piece is in a high transverse momentum jet and the other is in the underlying event.

In figure 1 we show a space–time picture of two string pieces stretched between two pairs of partons in this parallel frame. Since massless partons are propagating at the speed of light irrespective of the magnitude of their momenta, only the angles between them are important for the following. In the parallel frame the two string pieces have the same opening angle $\theta$, and the partons of one piece propagates with an angle $\theta/2$ w.r.t. the $z$-axis. The partons of the other propagates in the opposite direction, with an angle $\pi - \theta/2$. At any given time, both string pieces will lie in planes parallel to the $xy$-plane and to each other. Looking at the projections of the string pieces on the $xy$-plane, we denote the angle between them by $\phi$, and the frame is chosen such that all partons form an angle $\phi/2$ with the $x$-axis.

To simplify the calculations we write the momenta of the partons using their transverse momentum, $p_\perp$, and pseudo-rapidity difference, $\eta$, with respect to the $z$-axis, rather than the energy and opening polar angle (where $p_z = e \cos\frac{\theta}{2} = p_\perp \sinh\frac{\eta}{2}$), and get, using the notation

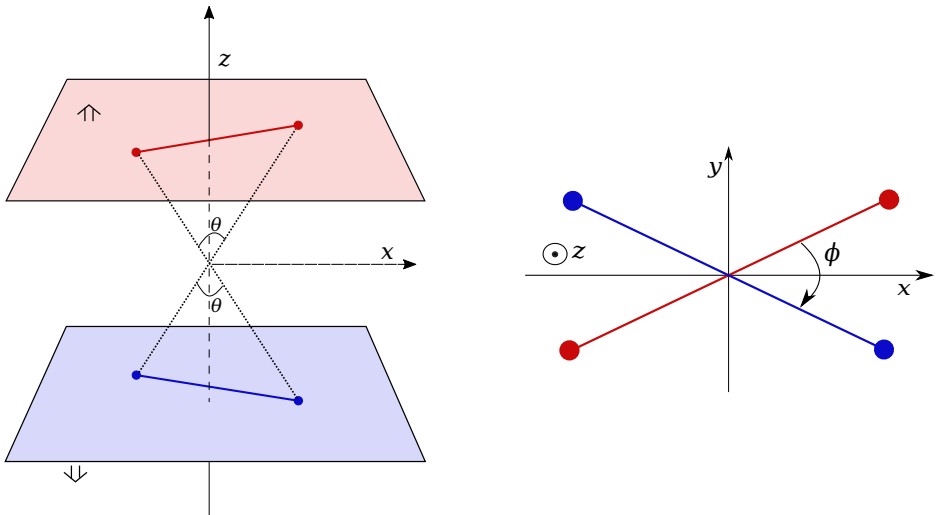

Figure 1: The parallel frame showing the parallel planes of two strings and the opening angle $\theta$ and skewness angle $\phi$.

$p = (e; p_x, p_y, p_z)$,

$$
\begin{aligned}
p_1 &= p_{\perp 1}\left(\cosh\frac{\eta}{2};\quad \cos\frac{\phi}{2},\quad \sin\frac{\phi}{2},\quad \sinh\frac{\eta}{2}\right), \\
p_2 &= p_{\perp 2}\left(\cosh\frac{\eta}{2}; -\cos\frac{\phi}{2}, -\sin\frac{\phi}{2},\quad \sinh\frac{\eta}{2}\right), \\
p_3 &= p_{\perp 3}\left(\cosh\frac{\eta}{2};\quad \cos\frac{\phi}{2}, -\sin\frac{\phi}{2}, -\sinh\frac{\eta}{2}\right), \\
p_4 &= p_{\perp 4}\left(\cosh\frac{\eta}{2}; -\cos\frac{\phi}{2},\quad \sin\frac{\phi}{2}, -\sinh\frac{\eta}{2}\right).
\end{aligned}
\tag{11}
$$

Clearly we have six degrees of freedom, and we can construct six independent squared invariant masses, $s_{ij} = (p_i + p_j)^2$. This means that for any set of four *massless* partons we can (as long as no two momenta are completely parallel) solve for $p_{\perp i}$, which will give us:

$$
p_{\perp 1}^2 = \frac{s_{12}}{4}\sqrt{\frac{s_{13}s_{14}}{s_{23}s_{24}}}, \quad
p_{\perp 2}^2 = \frac{s_{12}}{4}\sqrt{\frac{s_{23}s_{24}}{s_{13}s_{14}}}, \quad
p_{\perp 3}^2 = \frac{s_{34}}{4}\sqrt{\frac{s_{13}s_{23}}{s_{14}s_{24}}}, \quad
p_{\perp 4}^2 = \frac{s_{34}}{4}\sqrt{\frac{s_{14}s_{24}}{s_{13}s_{23}}},
\tag{12}
$$

and furthermore solve for the angles $\phi$ and $\eta$:

$$
\cosh\eta = \frac{s_{14}}{4p_{\perp 1}p_{\perp 4}} + \frac{s_{13}}{4p_{\perp 1}p_{\perp 3}} \quad \text{and} \quad \cos\phi = \frac{s_{14}}{4p_{\perp 1}p_{\perp 4}} - \frac{s_{13}}{4p_{\perp 1}p_{\perp 3}}.
\tag{13}
$$

To further specify the frame we renumber the particles so that $\phi < \pi/2$ to have the strings more parallel to the $x$-axis and not to the $y$-axis, and we define the $x$-axis to be their combined *rope* axis. The result is that for a breakup at a given space–time point in one string piece, we can in the parallel frame have a reasonable handle on the overlap with any other string piece.

## 3.2 Overlap in the parallel frame

In eq. (6) we wrote down the interaction energy of two completely parallel strings separated by a small distance. We now want to use this to estimate the effective overlap of two strings that are not completely parallel, but lie in parallel planes.

At a specific point along the $x$-axis in the parallel frame we denote the separation between the stings in the $yz$-plane by $(\delta_y, \delta_z)$ and integrate the interaction of the field given the skewness angle $\phi$ to obtain

$$
\begin{aligned}
\mathcal{I}(\delta_y, \delta_z, \phi) &= \int d^2\rho\, \mathbf{E}_1(\rho) \cdot \mathbf{E}_2(\rho) \\
&= E_0^2 \cos\phi \int dy\, dz \exp\left(-\frac{y^2 \cos\frac{\phi}{2} + z^2}{2R^2}\right) \exp\left(-\frac{(y-\delta_y)^2 \cos\frac{\phi}{2} + (z-\delta_z)^2}{2R^2}\right) \\
&= 2\pi E_0^2 R^2 \frac{\cos\phi}{\cos\frac{\phi}{2}} \exp\left(-\frac{\delta_y^2 \cos\frac{\phi}{2} + \delta_z^2}{4R^2}\right).
\end{aligned}
\tag{14}
$$

Here we note that the skewness angle enters both in the scalar product and in the strength of the field along the $y$-axis, and that the overlap vanishes for orthogonal strings.

We can now define the relative overlap as $\mathcal{I}(\delta_y, \delta_z, \phi)/\mathcal{I}(0,0,0)$ and use it as a probability (assuming that $\mathbf{E}_1 \cdot \mathbf{E}_2 > 0$) that a breakup in one string is affected by an increased string tension due to the overlap with the other. This would then correspond to a $\{2,0\} \to \{1,0\}$ transition giving an effective string tension $\kappa_{\text{eff}} = 3\kappa/2$ in eq. (10). If the strings instead points in the opposite directions along the $x$-axis ($\mathbf{E}_1 \cdot \mathbf{E}_2 < 0$) this would correspond to a $\{1,1\} \to \{0,1\}$ breakup with $\kappa_{\text{eff}} = 5\kappa/4$.

In this way we can for each breakup in one string piece, take all other string pieces in an event, and for each go to parallel frame to determine if it will contribute to $p$ or $q$. In our implementation described below, we sum the relative overlaps in $p$ and $q$ respectively and round them off to integers, rather than treating them as individual probabilities for each pair of string pieces, which on average gives the same result.

It should be pointed out that in the parallel frame we also have a handle on which string breaks up first. If we assume that the string breaks at a common average proper time along the string, $\tau_H$, we can in the parallel frame calculate the proper time of the other string in space–time point where we calculate the overlap. If the latter is larger $\tau_H$, we conclude that the other string has already broken up, and can no longer contribute to an increased string tension in the break-up being considered.

### 3.3 Monte Carlo implementation

The main technical problem with implementing the rope model in PYTHIA8, is the order in which the string fragmentation proceeds. First, the flavour and transverse momentum of the break-up is chosen (eq. (1)) together with the type of the chopped-off hadron. Only then is the momentum fraction, $z$, chosen according to eq. (2), and only then do we know exactly where the string breaks and can calculate the $\kappa_{\text{eff}}$ in that point. But we need to know $\kappa_{\text{eff}}$ to be able to calculate a break-up, so we have a kind of *Catch-22* situation.

The way we solve this is to perform a trial break-up to pre-sample the overlap of a given string, and use the overlap there to get an approximate $\kappa_{\text{eff}}$. Then we discard the sample break-up and produce a new one using this $\kappa_{\text{eff}}$. On the average we will then get a reasonable estimate of the overlap around a break-up. For a general break-up in the underlying event this should be good enough, but if we are interested in details of the hadron production in, *e.g.*, the tip of a jet, this procedure may be inappropriate (see further discussion below in section 4.3).

The procedure to calculate $\kappa_{\text{eff}}$ looks as follows:
1. Produce a trial break-up in the string being fragmented, and deduce from which string piece it comes.
2. Pair this piece with every other string piece in the event, make a Lorentz transformation to the parallel frame of each pair.

3. Using the pseudo-rapidity of the produced hadron in each such frame, and assuming the break-up occurred at the proper time, $\tau_H$, find the space-time point of the break-up of the first string piece.

4. In the corresponding $yz$-plane determine the proper time of the other string piece and if that is less that $\tau_H$, calculate the overlap according to eq. (14), and determine if this overlap should contribute to $p$ or $q$ in the breakup.

5. With the summed $p$ and $q$ (rounded off to integer values), we now calculate $\kappa_{\text{eff}}$ according to eq. (10).

6. Throw away the trial break-up with its produced hadron and change the PYTHIA8 fragmentation parameters according to the obtained $\kappa_{\text{eff}}$ and generate the final break-up.

As mentioned in section 2.1, some care has to be taken when it comes to soft gluons. Normally, all string pieces can be said to be *dipoles* between colour-connected partons, and in any parallel frame this string piece is parallel to the $xy$-plane. But a soft gluon may have lost all its momentum before the string breaks, and the break-up can then occur in a piece of the string that is not parallel to the string pieces of the connected dipoles. To include this possibility we introduce *secondary* dipoles, so that if we have two dipoles connected to a soft gluon, *e.g.* $q_i - g_j$ and $g_j - \bar{q}_k$, then a secondary string will be included spanned between the momenta of $q_i$ and $\bar{q}_k$, but using the space-time point where the gluon has lost all its momentum to the connected string pieces, as a point of origin.

The problem with soft gluons is present also for our shoving model in [15], and the solution with secondary dipoles is now also used there. This will be described in more detail in a future publication, where we also describe the procedure for including these *higher order* dipoles in cases where we have several consecutive soft gluons along a string.

### 3.4   Interplay with the Shoving model

Clearly our rope model is very tightly connected with our shoving model. They both rely on the parallel frame and technically they both use the same infrastructure for looking at overlaps between string pieces. However, here there is again a kind of *Catch-22*.

Physically the shoving precedes the hadronization, and pushes the strings apart before they hadronize. As this affects the value of $\kappa_{\text{eff}}$, the shoving should be executed first. However, for technical reasons the pushes are applied directly to the produced hadrons rather than to the individual string pieces. Therefore we must calculate the hadronization before we can execute the pushes.

We are currently working on a solution to this problem, and plan to present it in a future publication. The main effects of the shoving is expected to be a dilution of the strings resulting in a lowered $\kappa_{\text{eff}}$. As discussed in [15] the precise value of the string radius is not known, and in that paper we simply used a canonical value of 1 fm. Also the string radius will affect the values of $\kappa_{\text{eff}}$, and preliminary studies show that the effects of string dilution from shoving are of the same order as moderate decrease of the string radius of around 10%.

## 4   Results in pp collisions

In this section, features of the rope hadronization model with the parallel frame-formalism are investigated in pp collisions. Since the main feature of this new formalism is the much improved handling of string pieces which are not parallel to the beam axis (*i.e.* jets), we will mostly concentrate on observables in events containing a process with high momentum transfer, but in section 4.1 we first show the behaviour in minimum bias collisions. Here the most fundamental check of the dependence of $\kappa_{\text{eff}}$ with final state multiplicity, but more relevant

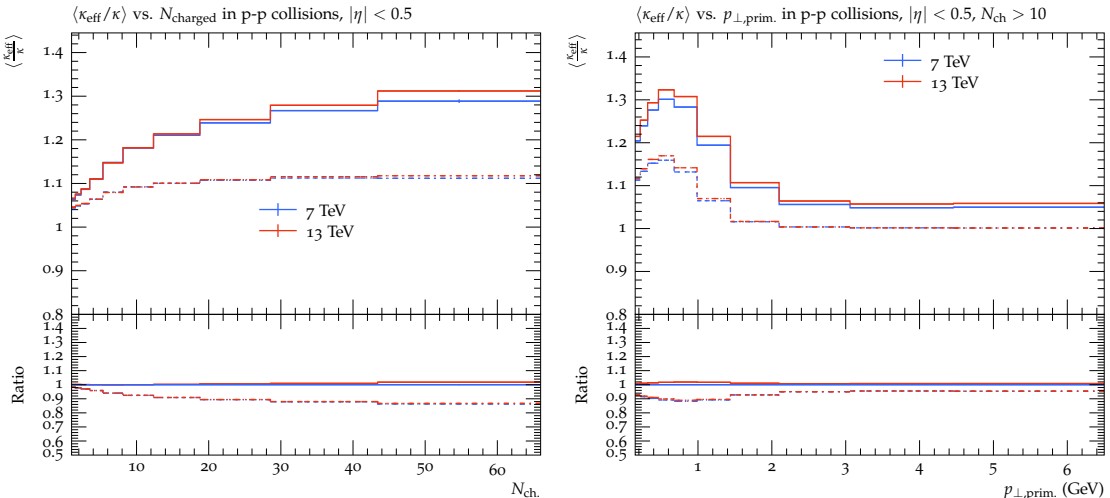

Figure 2: $\langle \kappa_{\mathrm{eff}}/\kappa \rangle$ vs. $N_{\mathrm{ch.}}$ (left) and vs. $p_{\perp,\mathrm{prim.}}$ for $N_{\mathrm{ch.}} > 10$ (right). Solid lines have string radius $R = 1$ fm and dot-dashed lines have $R = 0.5$ fm. Blue and red lines are for minimum-bias event at 7 and 13 TeV respectively.

for the parallel frame formalism, is the dependence of $\kappa_{\mathrm{eff}}$ on particle $p_\perp$. In section 4.2 we compare to existing experimental results in the underlying event (UE) for Z-triggered events. This is to ensure that the existing description of such observables is not altered by our model. Finally in section 4.3, we present predictions for the jet observables that are affected by rope formation in pp.

## 4.1   Model behaviour

In this section, we explore the variation of the effective string tension $\kappa_{\mathrm{eff}}$ with rope hadronization, for minimum bias pp events. The $\kappa_{\mathrm{eff}}$ is shown for primary hadrons, *i.e.* the effective string tension used to form a given hadron, produced directly in the hadronization process. Results are shown for two collision energies, $\sqrt{s} = 7$ and 13 TeV, and two values of string radius, $R = 0.5$ and 1 fm.

In figure 2, the dependence of $\langle \kappa_{\mathrm{eff}}/\kappa \rangle$ with respect to $N_{\mathrm{ch.}}$ in $|\eta| < 0.5$ is shown on the left, and $p_{\perp,\mathrm{prim.}}$ on the right. On the right, only events with $dN_{\mathrm{ch.}}/d\eta > 10$ are shown, to focus on events with several parton interactions. (At 13 TeV this corresponds to keeping roughly the 30% of events with the highest multiplicity [43].)

On the left plot of figure 2, it is seen that $\langle \kappa_{\mathrm{eff}}/\kappa \rangle$ rises with around 30% for $R = 1$ fm and 10% for $R = 0.5$ fm, almost irrespective of $\sqrt{s}$, with the rise at 13 TeV being only slightly higher. The two main points to take away from this figure, is a) that $dN_{\mathrm{ch.}}/d\eta$ is a good proxy for string density irrespective of collision energy, and thus works as a good scaling variable, and b) that any result will be very sensitive to the choice of $R$.

On the right plot of figure 2, we observe that the increase in $\kappa_{\mathrm{eff}}$ is larger for primary hadrons in the lower $p_\perp$ bins for both values of $R$. This means that the lower $p_\perp$ primary hadrons are formed from regions with high density of strings with more overlaps with adjacent strings. However, the higher $p_\perp$ partons correspond to "mini-jet" situations and are more separated in space-time from the bulk of strings. Such strings have less overlaps resulting in a lower $\kappa_{\mathrm{eff}}$. Hence the high $p_\perp$ primary hadrons formed from such string break-ups show this effect.

In the lowest $p_\perp$ bins of $\langle \kappa_{\mathrm{eff}}/\kappa \rangle$ vs $p_{\perp,\mathrm{prim.}}$ plot, it is seen that $\kappa_{\mathrm{eff}}$ drops to lower values. This behaviour arises from the fact that low $p_\perp$ particles are biased towards low $\kappa_{\mathrm{eff}}$ values due to the $p_\perp$-dependence on $\kappa$ in the tunneling probability in eq. (1).

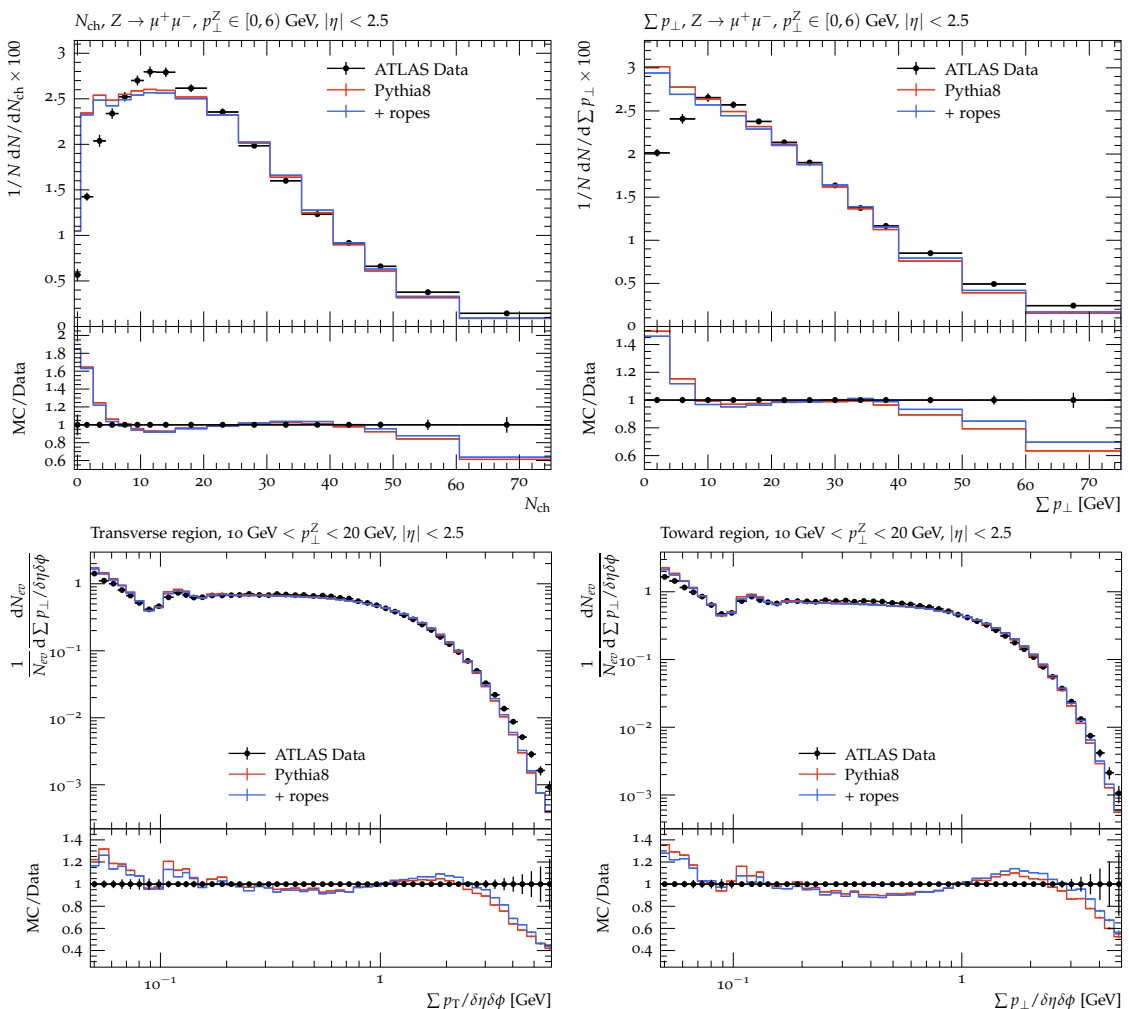

Figure 3: Associated particle production in Z→ $\ell^-\ell^+$ events at $\sqrt{s} = 7$ TeV compared to the default PYTHIA tune and with rope hadronization. **Top row:** Distribution of charged particle multiplicity, $N_{\text{ch}}$, (top left) and summed scalar transverse momenta, $\Sigma p_\perp$ (top right) measured for events with $p_\perp^Z$ range 0-6 GeV [44]. **Bottom row:** $\Sigma p_\perp$ distributions in different azimuthal regions, in events with $p_\perp^Z$ range 10-20 GeV [45]. Left: *transverse* region, $\pi/3 < |\Delta\phi_Z| < 2\pi/3$, right: *towards* region $|\Delta\phi_Z| < \pi/3$.

Overall we observe that rope hadronization significantly increase the string tension at high-multiplicities and for low $p_\perp$ final-state particles. For higher $p_\perp$ the effect is smaller, but does not disappear completely.

## 4.2 Underlying event observables in Z-triggered events

Before moving on to study rope effects on jets, it is important to assess whether rope formation drastically changes existing observables, currently well described by the existing model. In events with a Z-boson present, the most likely place for such a change to occur, is in the UE. To this end, we use a standard UE analyses implemented in the Rivet program [46].

In figure 3, $N_{\text{ch.}}$ and $\Sigma p_\perp$ for Z→ $\ell^-\ell^+$ events in pp collisions at 7 TeV are compared to ATLAS data [44,45]. The Z-boson is reconstructed from the electron or muon channel with invariant mass $66 < m_{\ell^-\ell^+} < 166$ GeV in $|\eta| < 2.5$.

The charged particle multiplicity and summed scalar $p_\perp$ distributions for Z→ $\mu^-\mu^+$ channel with $0 < p_\perp^Z < 6$ GeV, are shown in top row of figure 3. It is seen that adding rope hadroniza-

tion, overall preserves the distributions as produced by default PYTHIA8. We note that rope hadronization has a slight effect of pushing particles from lower to higher $\Sigma p_\perp$ regions, which follows from the $p_\perp$-dependence of the tunnelling probability in eq. (1).

The particle $p_\perp$ in the away region (opposite azimuthal region to that of the Z boson), balances the $p_\perp^Z$. Hence the towards and transverse regions with respect to the Z boson are much less affected by a recoiling jet and therefore have cleaner UE activity.[3] These regions are sensitive to the hadronization mechanism, rope hadronization effects will be apparent here. So we look at the UE-sensitive observables such as scalar summed $p_\perp/\delta\eta\delta\phi$ distributions for charged particles in events with $p_\perp^Z$ in the range 10-20 GeV in the bottom row of figure 3. These plots show the $\Sigma p_\perp$ distributions in the transverse ($\pi/3 < |\Delta\phi_Z| < 2\pi/3$) and towards ($|\Delta\phi_Z| < \pi/3$) regions [45]. We see that the rope hadronization curve follows the default PYTHIA8 curve, again preserving the overall physics behaviour of PYTHIA8, except for a slight shift in $\Sigma p_\perp$, as in the top right plot.

We conclude that UE measurements are equally well described with rope hadronization as without, and it is therefore not necessary to re-tune fragmentation parameters before proceeding to give predictions for jet observables.

## 4.3 Strangeness yields in Z+jet events

To investigate experimentally observable consequences of our rope model in terms of the yield of different hadron species inside jets, we have chosen to study its effects in Z+jets events at LHC energies. It has been shown in, *e.g.*, ref. [47], that such events are very useful for separating regions of phase space dominated by the UE from the regions dominated by jets. By selecting events where the Z boson is well balanced by a hard jet in the opposite azimuthal region, we can study the UE in a cone around the Z, where there should be very little activity related to the jet, and thus we can get a good estimate of the UE activity on an event-by-event basis. In this way we can get a reliable way of correcting jet observables for UE effects, not only for the transverse momentum of the jet but also for the flavour content.

### 4.3.1 Overall jet features

To observe the modification in the flavour production in the jet, we want to look at the yield ratios of different hadron species. Hence we have written a Rivet analysis where we first locate a reconstructed Z boson for $m_{\mu^-\mu^+}$ in the range 80-100 GeV and $|\eta| < 2.5$ and search for the hardest associated jet in the opposite azimuthal hemisphere. We further restrict the Z boson by requiring it to be within $|\eta| < 1.9$ and $p_\perp^Z > 8$ GeV using the standard Z-finding projection in Rivet. Once we find such a Z boson in the event, we search for the associated hardest (charged particle) jet using the anti-$k_T$ [48] algorithm with a radius $R_j = 0.4$ in $|\eta| < 2.1$ with the azimuthal separation $\Delta\phi_{\text{jet},Z} \geq 2\pi/3$.

To subtract UE contributions from the jet $p_\perp$, we calculate a characteristic $\Sigma p_{\perp,\text{UE}}$, by summing up the $p_\perp$ of the charged final state particles (not including muons from the Z decay) that lie within a cone of radius $\sqrt{2}R_j$ around the Z boson. Therefore, for a given event, the yields of the particles is calculated twice: once within the jet cone, then within a cone of radius $\sqrt{2}R_j$ with respect to the Z boson. The latter serves as our underlying event reference and we subtract half of this yield from the yield inside the jet cone to get the final yield of the hadrons in that event associated with the jet. Denoting the initial jet-$p_\perp$ as $p_{\perp,\text{pseudojet}}$, the corrected $p_{\perp,\text{jet}}$ becomes:

$$p_{\perp,\text{jet}} = p_{\perp,\text{pseudojet}} - 0.5 \times \Sigma p_{\perp,\text{UE}} \tag{15}$$

---

[3]It should here be noted that the underlying event activity in events with a hard interaction such as Z-production is generally higher than in minimum bias events.

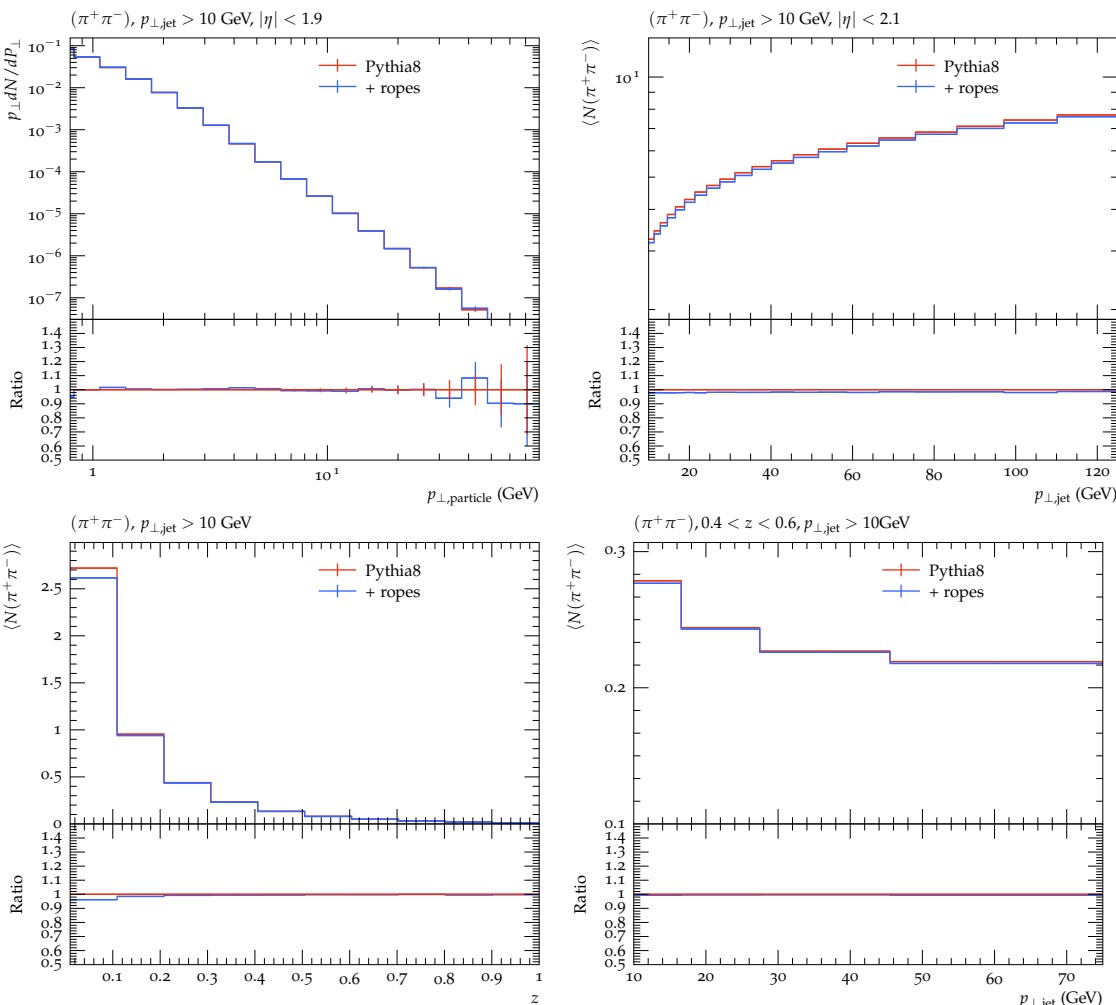

Figure 4: Pion yields in Z+jet events in 13 TeV pp collisions vs. $p_{\perp,\text{particle}}$ in the UE (top left), vs. $p_{\perp,\text{jet}}$ in the jet cone (top right), as a function of $z = p_{\perp,\text{particle}}/p_{\perp,\text{jet}}$ (bottom left), and vs. $p_{\perp,\text{jet}}$ for $0.4 < z < 0.6$ (bottom right).

and the corresponding yields:

$$\text{yield}_{\text{jet}} = \text{yield}_{\text{pseudojet}} - 0.5 \times \text{yield}_{\text{UE}} \tag{16}$$

This method of UE subtraction can easily be extended to p$A$ and $AA$ collisions to give a comparable result among the three systems. Similar methods have previously been used in heavy ion collisions [49]. We do this analysis for pp collisions at $\sqrt{s} = 13$ TeV with $p_{\perp,\text{jet}} \geq 10$ GeV for string radius $R = 1$ fm.

To examine the model performance in reproducing general features of the jets, such as particle multiplicity as a function of their transverse momentum and of the transverse momentum of the jets, we look at the pions. Our rope model is known to have very small effects on the overall multiplicity [25], and we know that pions in general are dominating the particle production, even though we expect a slight drop in pions, since high $\kappa_{\text{eff}}$ will favour strange hadrons and baryons over pions. In figure 4 we show the pion yield as a function of particle $p_\perp$ in the UE cone, and the UE-subtracted yield as a function of $p_{\perp,\text{jet}}$ in the jet cone in figure 4. We also show the pion yield with respect to $z = p_{\perp,\text{particle}}/p_{\perp,\text{jet}}$ and in the mid-$z$ region as a function of $p_{\perp,\text{jet}}$. Indeed we find that the rope effects are very small for pion production, both in the UE and in the jet, with the possible exception of the lowest bin in the $z$ distribution.

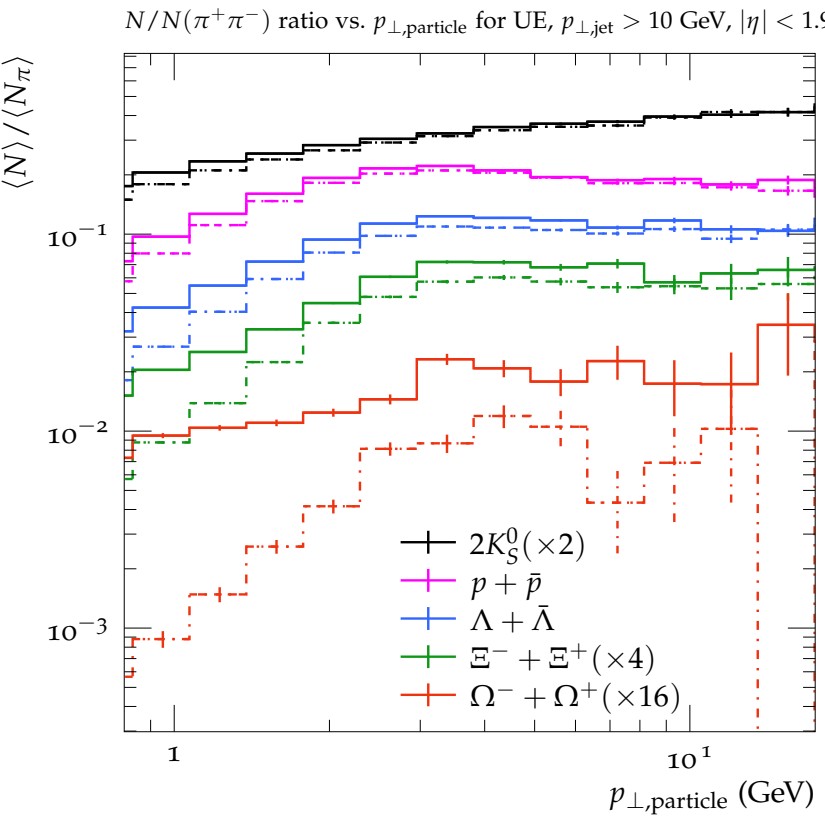

Figure 5: Yield ratio of different strange hadron species and protons to pions in the UE cone vs. $p_{\perp,\text{particle}}$, scaled by factors to show them clearly. Solid lines are with rope hadronization and dot-dashed lines are for default PYTHIA8.

We will revisit the bottom row plots in connection with strangeness yields in the jet cone in section 4.3.2.

In the UE region, the density of strings is high resulting in a higher number of overlaps among them. As a result, we would expect large effects due to rope hadronization in the UE. In order to observe this effect, we look at the yield ratio of the strange hadrons to pions in the UE cone. In figure 5, we show the yield ratio to pions for strange mesons ($K_S^0$) and baryons ($\Lambda$, $\Xi$ and $\Omega$) and protons with respect to $p_{\perp,\text{particle}}$. Yields of $\Xi$ and $\Omega$ baryons have been scaled by a multiplicative factor to show them in comparison to the other species. As expected, the different yields are higher with rope hadronization turned on as compared to default PYTHIA8. The highest enhancement for each species is observed for the lowest $p_{\perp,\text{particle}}$ ranges which subsequently decreases for higher particle $p_\perp$ (which follows figure 2 in section 4.1). Therefore, this plot show us that with rope hadronization, we get increased yields of baryons and strangeness. This plot also shows us the UE contribution to strangeness yields to that of within the jet.

Turning to flavour production *inside* the jet cone in figure 6, we show the UE-subtracted yield ratio to pions for the same set of hadron species as before, now with respect to $p_{\perp,\text{jet}}$. As rope hadronization will increase both strangeness and baryon production, the largest enhancement is expected for multistrange baryons. For $K_S^0$, only a slight increase is observed, while the increase for protons is higher. The $\Lambda$ yield due to rope hadronization is even higher due to combined baryon and strangeness enhancement. The yield of $\Xi$ is $\sim 20\%$ higher due to rope hadronization, than default PYTHIA8 and the $\Omega$ yield with rope hadronization is more than 50% higher. This shows that both baryon and strangeness yields are enhanced by rope

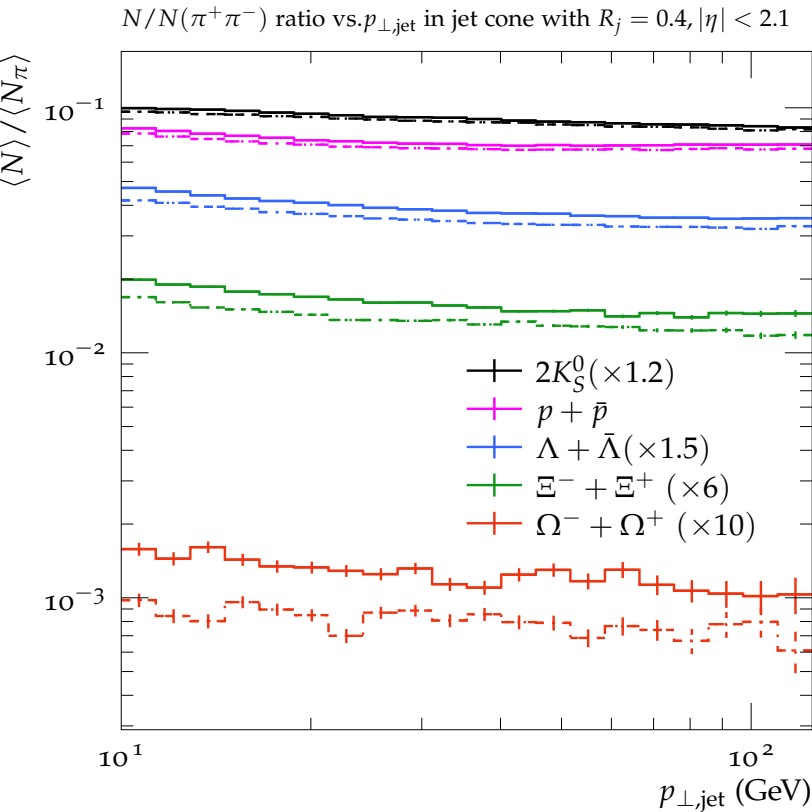

Figure 6: Yield ratio of different strange hadrons and protons to pions in the jet cone, for $R_j = 0.4$ vs. $p_{\perp,\text{jet}}$, scaled by factors to show them clearly. Solid lines are with rope hadronization and dot-dashed lines are for default PYTHIA8.

hadronization. We note that the increase in the yield ratio due to rope hadronization is rather constant over all $p_{\perp,\text{jet}}$. Hence if we look at the enhancement as a function of the transverse momentum ratio of the particle species to that of the jet, that would help us identify the $p_\perp$ ranges where rope effects are higher.

### 4.3.2 Jet substructure observables

Now we take a closer look at the particle to pion yield ratios as a function of $z$ and $p_{\perp,\text{jet}}$. Studies have been performed where the ratio of $p_\perp$ of the individual sub-jets to that of the leading jet serves as a distinguishing observable for jet modification [50]. Since we want to look at the strange flavour yields in the jet cone, we take a simpler approach. We only plot the yield ratios in bins of $z$, which is the ratio of the particle $p_\perp$ to the jet $p_\perp$.

In figure 7, we show the yield ratio of strange hadrons to pions *vs. z*. We observe that the particle yields are increased at low (close to the UE) to intermediate $z$ values. Furthermore, this enhancement is smaller for $K_S^0$ and larger for the strange baryon $\Lambda$, and for multistrange baryons $\Xi$ and $\Omega$ as expected. However, strangeness and baryon enhancement drops at higher $z$. This highlights the behaviour that rope hadronization effects decrease with higher $p_\perp$, as we noted in figure 2 in section 4.1.

We note that, even though the parallel frame formalism allows the calculation of $\kappa_{\text{eff}}$ in events with jets, the current implementation is lacking in the region $z \approx 1$, as already mentioned in section 3.3. The previously mentioned *Catch-22* situation, is purely related to the implementation, and can be further understood by considering the shape of the Lund symmetric fragmentation function in eq. (2), which is vanishing near $z = 1$. For a particle with $z$ close

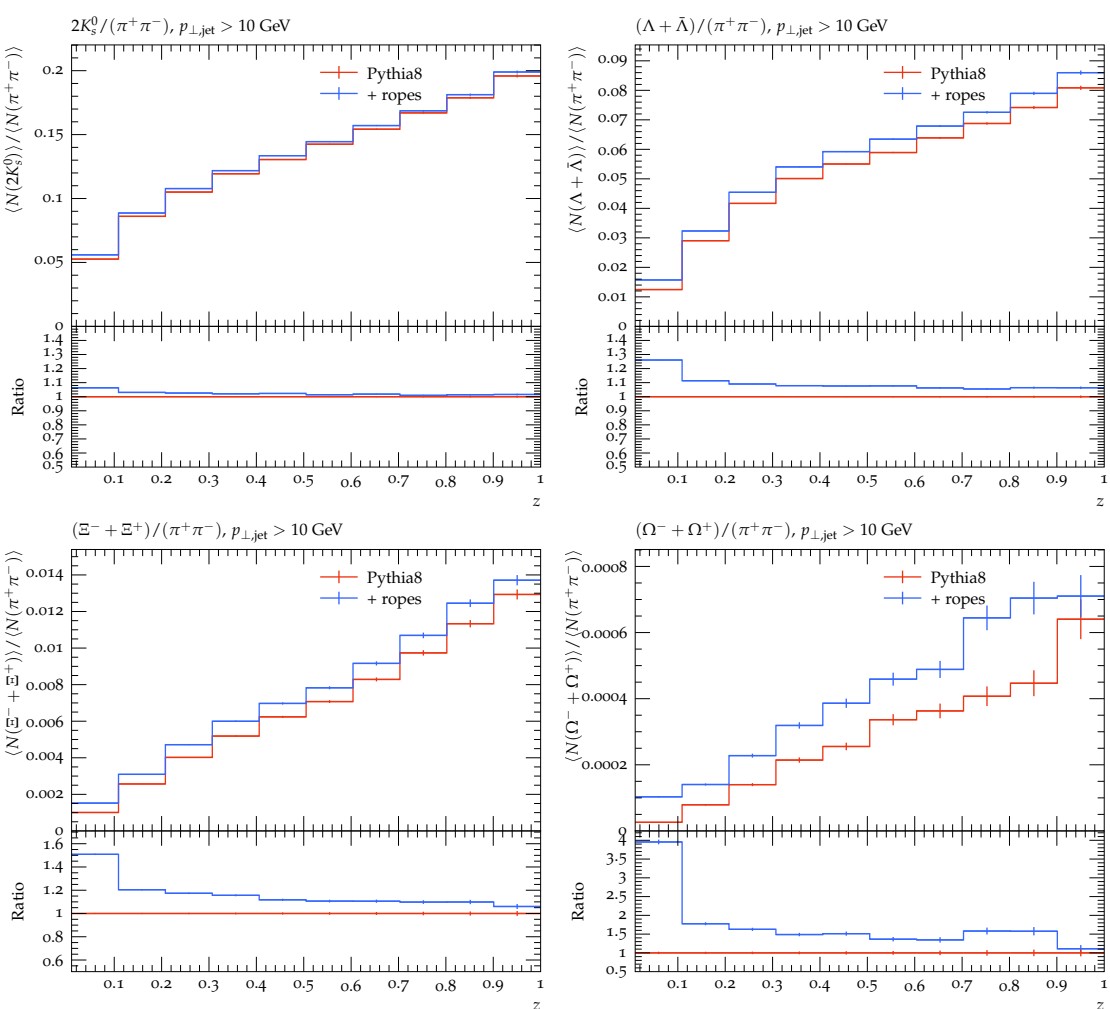

Figure 7: Yield ratios as a function of $z = p_{\perp,\text{particle}}/p_{\perp,\text{jet}}$ for pp collisions at $\sqrt{s}$=13 TeV: $2K_S^0/(\pi^+\pi^-)$ (top left) $(\Lambda + \bar{\Lambda})/(\pi^+\pi^-)$ (top right), $(\Omega^- + \Omega^+)/(\pi^+\pi^-)$ and $(\Xi^- + \Xi^+)/(\pi^+\pi^-)$ (bottom left).

to one, the pre-sampled overlap is therefore likely to have been calculated with a too-small $z$, which in turn means that it is calculated for the wrong part of the string. In pp collisions this effect is small but non-negligible, which we have confirmed by an *a posteriori* check (as the correct overlaps can be calculated after the fact, but too late to be used in event generation). Another issue, which would be present even in a perfect implementation, and therefore potentially more severe, is the absence of interactions between hadrons formed early in time, and their surrounding environment. For most of the produced particles, and in particular in pp, this effect should also be small. But in the case of high $z$, the particle is always produced early, and the effect could be larger. We plan to develop the model further in this direction, but in the meantime we will in the following show results for particles at intermediate $z$ values ($0.4 < z < 0.6$) where the effects arising from both these issues, should be negligible.

To test the modification in flavour yields at mid-$z$ values, we look at particle yields as a function of $p_{\perp,\text{jet}}$. Since these particles are neither close to the tip of the jet, nor to the UE, it is more reasonable to the trial-hadron sampling of $\kappa_{\text{eff}}$ in these regions. Moreover, as the jet $p_\perp$ increases, the particles get further and further away from the UE. In figure 8, we show the yield ratio of strange hadrons to pions in the $0.4 < z < 0.6$ region vs. $p_{\perp,\text{jet}}$. We observe that the yields from the rope hadronization case is distinct compared to default PYTHIA8. The individual

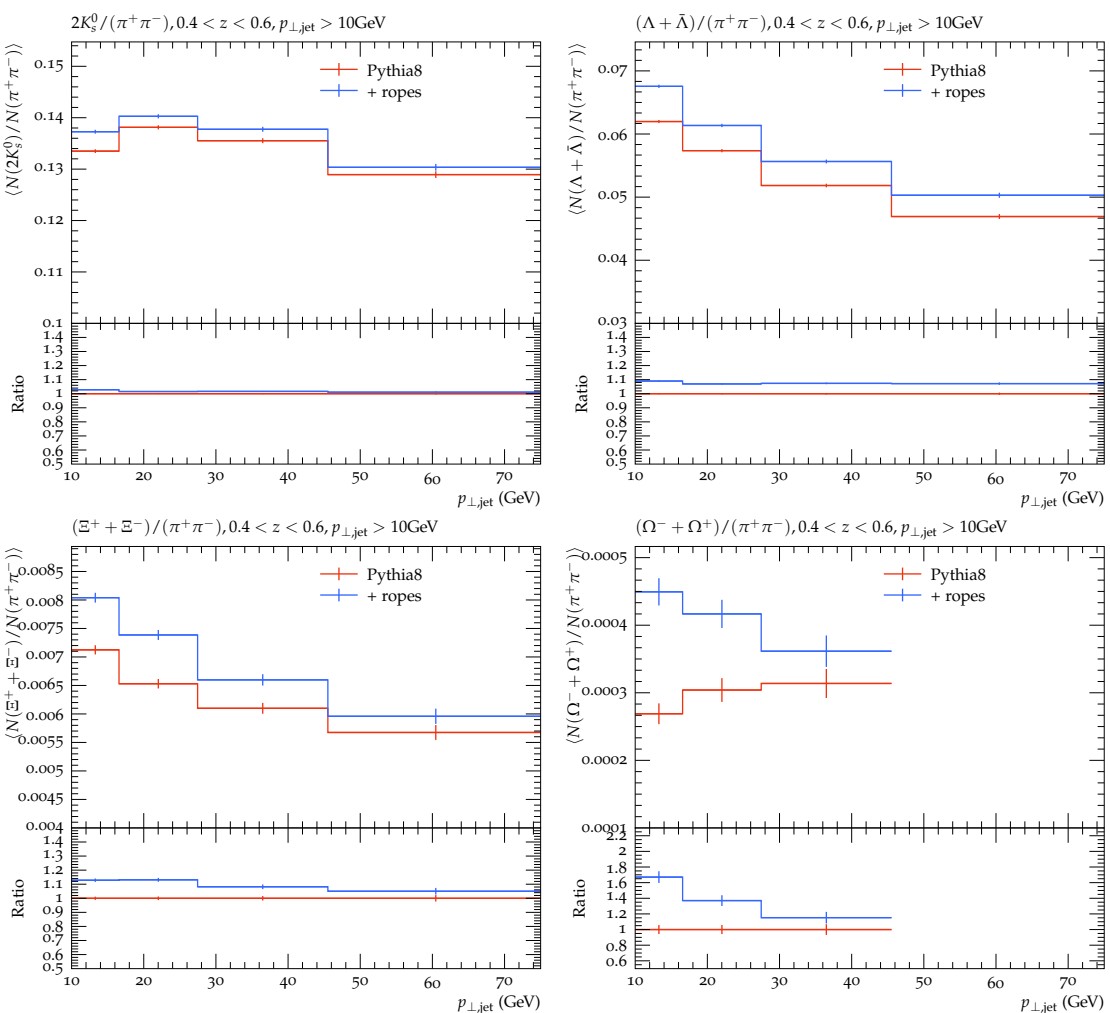

Figure 8: Yield ratios of particles with $0.4 < z < 0.6$, as a function of $p_{\perp,\text{jet}}$ for pp collisions at $\sqrt{s}$=13 TeV: $2K_s^0/(\pi^+\pi^-)$ (top left), $(\Lambda + \bar{\Lambda})/(\pi^+\pi^-)$ (top right), $(\Xi^- + \Xi^+)/(\pi^+\pi^-)$ (bottom left) and $(\Omega^- + \Omega^+)/(\pi^+\pi^-)$ (bottom right).

strange hadron yield to pion yield ratio increases as we go from the $K_s^0$ meson to the $\Lambda$ baryon (top row plots). For multistrange baryons, $\Xi$ and $\Omega^-$ (bottom row plots), rope effects are amplified due to higher number of strange quarks, resulting in a 20% - 50% increase in their yields in low $p_{\perp,\text{jet}}$ ranges. However, as mentioned before, we would expect the enhancement in the yields to drop at higher $p_{\perp,\text{jet}}$ bins. This effect is rather small for $\Lambda$ but prominent for $\Xi$ and $\Omega$. $\Omega$ (bottom right plot) is only shown up to 45 GeV due to statistics.

## 5  Conclusion

We have here presented a study on how an effect from a dense system of colour fluxtubes might be observed as strangeness enhancement in jets in high multiplicity pp events. In such events it is essential to properly estimate the interaction between non-parallel strings, including strings connected to a hard scattered parton and strings in the underlying event. This problem was solved in ref. [15], where the interaction of all string pairs can be calculated in a Lorentz frame, where two string pieces lie symmetrically in two parallel planes. We here show results for jet-triggered high-multiplicity pp collisions. The generalization to p*A* and *AA* collisions (using the

Angantyr model [51]) will be presented in a future publication.

The interacting strings can form "colour ropes", which hadronize in a stepwise manner by $q\bar{q}$ pair creation. The increased energy in the rope gives a higher "effective string tension", $\kappa_{\text{eff}}$, which increases the number of strange quarks and diquarks in the breakups. In section 4.1 we found that this results in an increase of $\kappa_{\text{eff}}$ with multiplicity in pp events at LHC energies. It is interesting to note that the increase for a given multiplicity is almost independent of the collision energy.

As expected we also found that the increase is quite dependent on the transverse momentum, since high-$p_\perp$ particles are typically produced in jets where the strings are not parallel with the bulk of the strings in the underlying event, thus reducing the effective overlap with these. The important question is then if the rope model, despite being reduced in jets, anyway will result in a modification of the hadron composition of jets.

To study the effects on jets we focused our investigation on Z+jet events, with the Z decaying to lepton pairs. As pointed out in *e.g.* ref. [47], it is possible, in such events, to get a relatively clean separation between the jets and the particle production in the underlying event. In particular the hadrons produced in a cone around the direction of the Z particle should have very little to do with the recoiling jet, and can therefore be used to correct any observable in the jet cone for underlying-event contributions on an event-by-event basis.

The modified $\kappa_{\text{eff}}$ also affects the fragmentation parameters. In section 4.2 results for multiplicity and the transverse momentum distribution in the underlying event in pp Z+jet events, were compared with results from default PYTHIA8 and with data from ATLAS. After confirming that the rope hadronization gives negligible effects on these general features of the underlying event, we feel comfortable that we can study strangeness and baryon enhancement in the jets in a way, which is not biased by the underlying-event corrections.

In section 4.3 our main results for strangeness and baryon number enhancement in jets were presented, with the underlying event subtracted. We note that the effect is most important for strange baryons, and growing with the number of strange quarks. Thus it is largest for $\Omega$ baryons, and from the plots showing the $\Omega/\pi$ ratio as a function of the jet transverse momentum, we note that rope effects are very small for large jet $p_\perp$ as expected, but quite noticeable for low jet $p_\perp$.

From this we conclude that it may indeed be possible find jet modifications due to collective effects, in our rope model, in small collision systems. The size of the effect is, however, a bit uncertain. In part this is due the uncertainty in the transverse size of the string, and our canonical choice of $R = 1$ fm may be a bit large. Although it should be possible to tune this parameter to fit the overall strangeness and baryon enhancement, it is then also important to also take into account the effects of repulsion between the strings. Both of these effects will be addressed in future publications.

Looking ahead, it is also interesting investigate the effects of colour reconnection, in particular models that include junction formations, which will also influence the baryon production. In the end we hope to develop a picture where most collective effects can be interpreted as interactions among strings, not only in pp collisions but also in p*A* and *AA*.

# Acknowledgements

This work was funded in part by the Knut and Alice Wallenberg foundation, contract number 2017.0036, Swedish Research Council, contracts number 2016-03291, 2016-05996 and 2017-0034, in part by the European Research Council (ERC) under the European Union's Horizon 2020 research and innovation programme, grant agreement No 668679, and in part by the MCnetITN3 H2020 Marie Curie Initial Training Network, contract 722104.

# A   Dependence of fragmentation parameters on $\kappa_{\text{eff}}$

There are several hadronization parameters in PYTHIA8, and even if they are in principle independent, several of them has an implicit dependence on the string tension. In our implementation of the rope hadronization, we take the parameters as tuned to $e^+e^-$ data, where we expect no rope effects, and for each breakup in the string fragmentation we rescale the parameters according to the estimated change in string tension at that point, due to the presence of overlapping string fields. The parameters under consideration is the same as in our previous implementation [25], and the dependence of the string tension is also the same. For completeness we list them here, but for further details we refer to [25].

In the following we will denote the change in string tension by $h$, according to $\kappa \mapsto \kappa_{\text{eff}} = h\kappa$. The following parameters is affected:

- $\rho$ (`StringFlav:probStoUD`[4]): the suppression of $s$ quark production relative to $u$ or $d$ type production. This parameter has a simple scaling

$$\rho \mapsto \tilde{\rho} = \rho^{1/h}. \tag{17}$$

- $x$ (`StringFlav:probSQtoQQ`): the suppression of diquarks with strange quark content relative to diquarks without strange quarks (in addition to the factor $\rho$ for each extra $s$-quark) also scales like

$$x \mapsto \tilde{x} = x^{1/h}. \tag{18}$$

- $y$ (`StringFlav:probQQ1toQQ0`): the suppression of spin 1 diquarks relative to spin 0 diquarks (not counting a factor three due to the number of spin states of spin 1 diquarks) again scales like

$$y \mapsto \tilde{y} = y^{1/h}. \tag{19}$$

- $\sigma$ (`StringPT:sigma`): the width of the transverse momentum distribution in string break-ups. This is directly proportional to $\sqrt{\kappa}$, giving

$$\sigma \mapsto \tilde{\sigma} = \sigma \sqrt{h}. \tag{20}$$

- $\xi$ (`StringFlav:probQQtoQ`): the global probability of having a diquark break-up relative to a simple quark break-up. This has a somewhat more complicated $\kappa$ dependence and also have uncertainties related to the so-called popcorn model as described in [25]. We decompose it as three different parameters, $\xi = \alpha\beta\gamma$ with different $\kappa$-dependence, where $\beta$ is related to the probability to have a $q\bar{q}$ fluctuation in general in the popcorn model which is independent of $\kappa$ and is treated as an independent parameter, while $\gamma$ is related to the masses and scales as

$$\gamma \mapsto \tilde{\gamma} = \gamma^{1/h}, \tag{21}$$

and $\alpha$ is related to the different di-quark states with an indirect dependence on $\rho$, $x$, and $y$

$$\alpha \mapsto \tilde{\alpha} = \frac{1 + 2\tilde{x}\tilde{\rho} + 9\tilde{y} + 6\tilde{x}\tilde{\rho}y + 3\tilde{y}\tilde{x}^2\tilde{\rho}^2}{2 + \tilde{\rho}}. \tag{22}$$

Taken together we get the following dependence:

$$\xi = \alpha\beta\gamma \mapsto \tilde{\xi} = \tilde{\alpha}\beta \left( \frac{\xi}{\alpha\beta} \right)^{1/h}. \tag{23}$$

---

[4]This is the parameter name in PYTHIA8.

- $b$ (`StringZ:bLund`): the parameter in the symmetric fragmentation function eq. (2) scales with the $\rho$-parameter as follows

$$b \mapsto \tilde{b} = \frac{2 + \tilde{\rho}}{2 + \rho}\, b. \tag{24}$$

- $a$ (`StringZ:aLund`): the other parameter in eq. (2) has an indirect dependence on $b$ through the normalisation, $N$, in eqs. (**??**) and (2). This does not have a simple analytic form, so the scaling of $a \mapsto \tilde{a}$, is instead done with a numeric integration procedure.

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
