# Peer review of "Jet modifications from colour rope formation in dense systems of non-parallel strings"

_SciPost Physics Core_

## Round 1 · Referee Report · Anonymous · 2022-4-21

Strengths
1. Clearly written
2. Good model explanation
3. Good choice of observables to highlight effect
Report
The authors report on a model to describe the formation of colour ropes in dense systems of colour strings which are formed in a hadronic collision. The key ingredients of the model implementation are the string overlap, computed in the so-called parallel frame and an alteration of the string tension to an effective value $\kappa_{\rm eff}$. Both allow to use the ordinary Pythia string fragementation model, but with modified parameters to reflect the effect of the density of strings. The most important effect to expect is that the suppression of heavy quarks, mostly strange quarks, relative to light up and down quarks in the string breakup is relaxed by a higher string tension, leading to an enhancement of strange particle production in dense systems.
The model is explained very clearly in relation to the existing string fragmentation model and at various places the phenomenological implications are already pointed out. The authors focus a study of the effect of the model to pp collisions first and leave a more involved study with ions to a later publication. In order to single out the implications for jets, which can be studied well in Z+jet events the effects on the underlying event in general are shown to be small. Moving to the jet that recoils against the Z-boson, the observed effects can be assumed to stem from the high particle density within the jet itself. It is shown that strange particle production is enhanced within the jet, particularly for lower transverse momenta of the jet itself and for lower momentum fractions of the considered hadrons.
The paper documents an important step towards a microscopic description of high-density hadronic systems and it will be interesting to see future results for the more complex pA and AA collisions.
For convenience I point out a few typos that have been spotted while reading the manuscript:
p7, next-to last paragraph: "propagates -> propagate" (twice)
p9, 2nd line: "stings" -> "strings"
"to parallel frame" -> "to the parallel frame"
"the latter is larger" -> "the latter is at larger"
p10, bullet 4, "less that" -> "less than"
p10, last paragraph of 3.4: "main effects" -> "main effect"
p12, first line, "increase" -> "increases"
p14, end of 2nd paragraph: "show" -> "shows"
p17, last line "case is" -> "case are"
p19, next-to last paragraph: "possible find" -> "possible to find"
p19, last paragraph: "interesting investigate" -> interesting to
investigate"
p20, last bullet: "have uncertainties" -> "has uncertainties"
p21, near end: one equation reference is unresolved.
Requested changes
1- In the introduction a number of possibilities to address the enhancement of strange particle production are addressed. A note on the result of Herwig's Baryonic Colour Reconnection model might suit here as well, as this also enhances strange particle production in dense systems.
2- One small confusion may be easily resolved. The authors mention a flux tube radius of 0.5 fm after Eq. (4) and an unknown string radius, assumed to be 1 fm. Aren't these in any way related and would it matter at all to choose different values for these?
Deepak Kar on 2022-03-20 [id 2308]
Dear authors,
I read with interest this paper on rope hadronisation, and congratulations on a thorough result. As an experimental (that too pp, not HI) physicist, I mostly focussed on Sec.4, to see what it buys us in terms of improved modelling or a better estimation of uncertainties. This is not an invited report :).
Section 4.2: I think I dont disagree too strongly that ZUE at 7 TeV is a good indicator of the robustness, but perhaps 13 TV lead track UE would have been extra reassuring.
Minor correction: I know what authors mean, but we use trigger in a specific sense, so these are single-muon triggered events!
Sec 4.3: An alternative approach to look at clean UE was proposed in https://arxiv.org/abs/1801.05218 , and used in https://arxiv.org/abs/1905.09752 ;-).
I am not sure I fully understand the motivation of sqrt(2)R_j, could not this correction be done with ghost area?
Minor: footnote 3, you mean charged particle activity, UE is only when there is a hard probe.
Generally, it is good to see the difference that this new model produces in the distributions shown, but would have been useful to see some comparisons with data, such as
ALICE_2015_I1357424
ALICE_2014_I1300380
ATLAS_2011_I944826
CMS_2011_S8978280
LHCB_2011_I917009
CMS_2012_PAS_QCD_11_010
And finally, a question to the authors. We are planning a quick minbias-y measurement with the upcoming low-mu 13.6 TeV data in ATLAS. While the marginal increase in the sqrt(s) makes the usual MB observables not so exciting, if some new simple track-based observables can be suggested to test these models, we can consider that!
Best wishes,
Deepak

---

## Editorial Decision

submission_&_refereeing_history